# Accessing chiral sulfones bearing quaternary carbon stereocenters via photoinduced radical sulfur dioxide insertion and Truce–Smiles rearrangement

Jiapian Huang [1,5], Fei Liu [1,5], Ling-Hui Zeng [2], Shaoyu Li [1] ✉,
Zhiyuan Chen [2] ✉ & Jie Wu [1,3,4] ✉

From the viewpoint of synthetic accessibility and functional group compatibility, photoredox-catalyzed sulfur dioxide insertion strategy enables in situ generation of functionalized sulfonyl radicals from easily accessible starting materials under mild conditions, thereby conferring broader application potential. Here we present two complementary photoinduced sulfur dioxide insertion systems to trigger radical asymmetric Truce–Smiles rearrangements for preparing a variety of chiral sulfones that bear a quaternary carbon stereocenter. This protocol features broad substrate scope and excellent stereospecificity. Aside from scalability, the introduction of a quaternary carbon stereocenter at position β to bioactive molecule-derived sulfones further demonstrates the practicality and potential of this methodology.

As privileged structural units, chiral sulfones are pervasive in a variety of biologically active products and pharmaceutical molecules whose function could span from CXCR2 antagonist, antihypertensive renin inhibitor, r-secretase inhibitor to antibacterial and antispasmodic activities (Fig. 1a)[1–7]. In synthetic development, these sulfone motifs permit versatile installation of other useful functionalities through directional derivatization such as alkylation, halogenation, nucleophilic substitution, oxidation, Julia olefination and Ramberg–Bäcklund reaction[8–11]. Owing to their established significance, the asymmetric synthesis of chiral sulfones has garnered considerable attentions from the synthetic chemistry community[12–16]. The renewed recent interest in radical chemistry due to their high reactivity and chemoselectivity has yielded more efficient radical-based asymmetric transformations. This development reveals new possibilities to directly prepare chiral sulfones from sulfonyl radicals with asymmetric synthetic protocols. In this respect, Lin and Liu have independently described copper-catalyzed enantioselective sulfonyl radical-initiated difunctionalization of alkenes to obtain a series of

chiral sulfone derivatives[17,18]. This approach benefits further from the emergence of visible light photoredox catalysis, which has enabled generation of radicals under mild conditions via single-electron transfer from precursors[19–21] or through radical insertion of sulfur dioxide[22–36]. Using a photocatalytic system comprised of chiral biscyclometalated rhodium(III) complex, Meggers and Wu accomplished enantioselective conjugate addition of sulfonyl radicals to activated alkenes for the efficient synthesis of chiral sulfones[19,20]. An elegant advance in photoinduced asymmetric sulfonylation was introduced by Gong and co-workers in 2021, which merges an inert C(sp[3])-H functionalization with sulfur dioxide insertion mechanism to form the sulfonyl radicals[33]. Very recently, we developed a photoinduced, organocatalytic asymmetric radical approach to access sulfonyl-containing axially chiral styrenes through a three-component reaction of potassium alkyltrifluoroborates, potassium metabisulfite, and 1-(arylethynyl)naphthalen-2-ols, in which alkylsulfonyl radicals are formed in situ from a sulfur dioxide insertion process[36]. These advances notwithstanding, photoinduced radical

[1]School of Pharmaceutical and Chemical Engineering &Institute for Advanced Studies, Taizhou University, Taizhou 318000, China. [2]School of Medicine, Zhejiang University City College, Hangzhou 310015, China. [3]State Key Laboratory of Organometallic Chemistry, Shanghai Institute of Organic Chemistry, Chinese Academy of Sciences, Shanghai 200032, China. [4]School of Chemistry and Chemical Engineering, Henan Normal University, Xinxiang 453007, China. [5]These authors contributed equally: Jiapian Huang, Fei Liu. ✉e-mail: lisy@tzc.edu.cn; chenzy@zucc.edu.cn; jie_wu@fudan.edu.cn

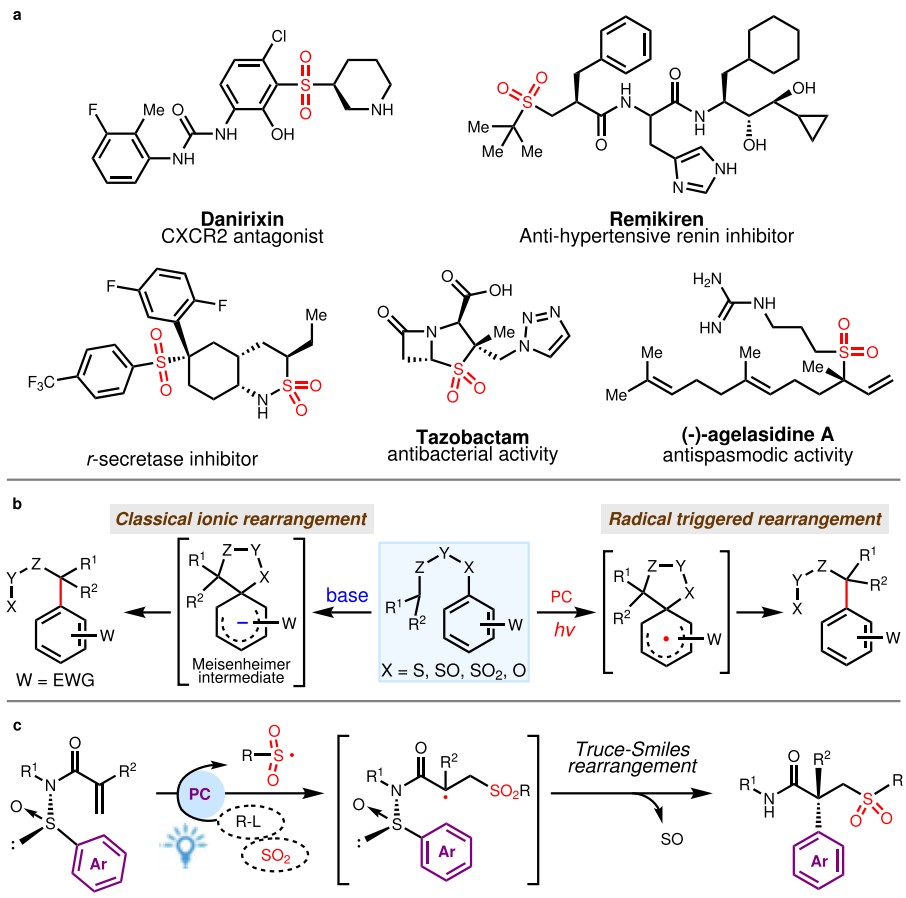

**Fig. 1 | Valuable chiral sulfone-containing molecules and their construction.**
**a** Representative bioactive molecules containing chiral sulfone motif. **b** Truce–Smiles rearrangement. **c** This work: radical sulfur dioxide insertion triggered asymmetric Truce–Smiles rearrangement.

method that could assemble chiral sulfones with quaternary carbon stereocenters asymmetrically remains elusive and challenging.

Truce–Smiles rearrangement reactions which occur by predictable cleavage and reconstruction of chemical bonds in facilitating an aryl group migration has unmatched utility in furnishing quaternary carbon centers. Classical ionic rearrangements could exhibit high electronic and steric dependencies, requiring special substrates with electron withdrawing groups as well as forcing conditions including strong base and high temperature to effect an efficient *ipso* nucleophilic substitution (Fig. 1b, left)[37,38]. As a complementary reaction mode capable of compensating for the deficiencies associated with the ionic pathway, radical-triggered Truce–Smiles rearrangement (Fig. 1b, right) has come to the forefront since Speckamp's pioneering work[39–42]. In parallel with the revival of photoredox chemistry, great strides were made in the radical paradigm of this reaction with substantial contributions from Stephenson, Studer, Sapi, Zhu, Nevado and others[43–54]. However, the low conversion energy has restricted the development of the asymmetric versions of this radical rearrangement. Only recently have Nevado and colleagues achieved the first asymmetric Truce-Smile rearrangement which generates a quaternary carbon stereocenter by designing a sulfoxide group as traceless chiral auxiliary on the substrates[54]. By using sulfonyl radicals generated from sulfonyl chlorides as initiators, this seminal protocol could be extended to construct chiral sulfones. Grounded in our long-term interest in sulfur dioxide insertion strategy[34–36], we were drawn to integrate this chemistry in a radical Truce–Smiles rearrangement which would constitute a strategic complement to this precedent. This motivation is particularly relevant considering the potentially tedious procurement of sulfonyl chlorides by chlorination of sulfonic acid or Sandmeyer reaction.

Moreover, they could exhibit poor compatibility with amino and hydroxyl groups as well as other sensitive motifs.

Here, we show our efforts in developing asymmetric three-component Truce–Smiles rearrangement reaction of chiral acrylamides, carbon radical precursors and sulfur dioxide surrogates (Fig. 1c). Two complementary photoinduced sulfur dioxide insertion systems are established to trigger radical asymmetric Truce–Smiles rearrangements, enabling the installation of various functionalized sulfonyl groups to the β-position of all-carbon quaternary stereocenters.

## Results

### Reaction condition optimization

In view of its unique potential as single-electron reductant[55], sodium dithionite was employed as the sulfur dioxide surrogate in combination with diazonium salts in our initial design to generate sulfonyl radicals for initiating the rearrangement. Using *N*-arylsulfinyl acrylamide **1a** and aryldiazonium salt **2a** as model substrates, we commenced the search for suitable reaction conditions with the corresponding results outlined in Table 1. The desired product **4a** could be afforded in 57% yield with 95% ee, by employing *fac*-Ir(ppy)$_3$ as the photosensitizer and CH$_3$CN as the solvent under the irradiation of 35 W blue LED at 20 °C for 18 h (entry 1). Further screening of photosensitizers revealed Mes-AcrClO$_4$ with excellent oxidizing capability to be the optimal candidate (entries 2–4). Other solvents including DMF, THF, DCE, and PhCF$_3$ did not yield further enhancement of this outcome (entries 5–8). By reasoning that a protonation should precede the product formation and Lewis basic acceptor would aid to intercept a departing SO cation, a series of additives capable of

## Table 1 | Optimization of the reaction conditions[a]

| Entry | PC | Solvent | Additive | Yield (%)[b] | Ee (%)[c] |
|---|---|---|---|---|---|
| 1 | fac-Ir(ppy)$_3$ | CH$_3$CN | / | 57 | 95 |
| 2 | Ir[dF(CF$_3$)ppy]$_2$(bpy)PF$_6$ | CH$_3$CN | / | 49 | 95 |
| 3 | 4CzIPN | CH$_3$CN | / | 51 | 95 |
| 4 | Mes-AcrClO$_4$ | CH$_3$CN | / | 62 | 96 |
| 5 | Mes-AcrClO$_4$ | DMF | / | ND | / |
| 6 | Mes-AcrClO$_4$ | THF | / | 42 | 96 |
| 7 | Mes-AcrClO$_4$ | DCE | / | 31 | 96 |
| 8 | Mes-AcrClO$_4$ | PhCF$_3$ | / | 10 | 96 |
| 9 | Mes-AcrClO$_4$ | CH$_3$CN | H$_2$O | 40 | 95 |
| 10 | Mes-AcrClO$_4$ | CH$_3$CN | MeOH | 34 | 94 |
| 11 | Mes-AcrClO$_4$ | CH$_3$CN | PhCO$_2$H | 68 | 96 |
| 12 | Mes-AcrClO$_4$ | CH$_3$CN | NaHSO$_3$ | 82 | 96 |
| 13[d] | Mes-AcrClO$_4$ | CH$_3$CN | NaHSO$_3$ | 40 | 95 |
| 14[e] | Mes-AcrClO$_4$ | CH$_3$CN | NaHSO$_3$ | 69 | 94 |
| 15[f] | Mes-AcrClO$_4$ | CH$_3$CN | NaHSO$_3$ | 31 | 94 |
| 16 | / | CH$_3$CN | NaHSO$_3$ | 50 | 95 |
| 17[g] | Mes-AcrClO$_4$ | CH$_3$CN | NaHSO$_3$ | 40 | 96 |

PC: Photocatalyst.
[a]Conditions: 1a (0.2 mmol), 2a (0.4 mmol), Na$_2$S$_2$O$_4$ (0.4 mmol), photocatalyst (2 mol%), additive (1.5 equiv), solvent (3.0 mL), 35 W blue LED, under N$_2$ at 20 °C for 18 h.
[b]Determined by 1H NMR analysis using 1,3,5-trimethoxybenzene as an internal standard.
[c]Determined by HPLC analysis.
[d]At 10 °C.
[e]At 30 °C.
[f]In the absence of Na$_2$S$_2$O$_4$.
[g]In the absence of visible light.

donating both proton and electron pair were introduced into the model reaction system. The addition of water, methanol and benzoic acid failed to bring about a notable positive effect in terms of product yield (entries 9–11) while sodium bisulfite significantly raised the product yield to 82% (entry 12). At this stage, the possibility of the latter to serve also as sulfur dioxide surrogate has yet to come to light. Performing the reaction at 20 °C was found to be most appropriate as a decreased or elevated reaction temperature led to notable erosion of yield to different extents (entries 13 and 14). Removal of $Na_2S_2O_4$ resulted in a dramatic drop in yield, showing that both $Na_2S_2O_4$ and $NaHSO_3$ were necessary (entry 15). Similarly, a decreased was observed in the absence of either Mes-AcrClO$_4$ or blue LED, suggesting that photosensitizer and visible light are crucial components for an efficient transformation (entries 16 and 17).

### Substrate scope

Having established the optimal reaction parameters for the envisioned Truce–Smiles rearrangement, the general applicability of this set of conditions was then evaluated. As displayed in Fig. 2, various aryldiazonium salts were smoothly transformed into the desired chiral sulfones with high retention of optical purities and a broad tolerance of functional groups. For instance, substrates bearing electron-rich (alkyl, MeO, MeS) or electron-deficient ($CF_3$, CN, $NO_2$, halogen) para-substituent invariably participated smoothly to afford moderate-to-good product yields and excellent stereoselectivities, suggesting that the electronic characteristics and substitution patterns had limited effect on the rearrangement reaction (**4a**–**4i**). Moreover, 3-bromo-substituted aryl sulfone **4j** was smoothly produced in 61% yield and 94% ee, indicating opportunity to introduce other functionalities onto product scaffold through metal catalyzed cross-coupling reactions. By contrast, 3-benzyloxy-substituted substrate **2** underwent the rearrangement with lower efficiency (**4k**). It is worthy of note that structurally more varied diazonium salts derived from benzofuran, coumarin and naphthalene were compatible with the photoredox catalytic system as well (**4l**–**4n**). The modulation of aryl ring within the N-arylsulfinyl acrylamide component was also feasible, furnishing chrial sulfones **4o**–**4t** in reasonable yields. The absolute configuration of compound **4p** was determined as R by X-ray crystallographic analysis (CCDC: 2208406) and those of other products were assigned by analogy. It was further demonstrated that the excellent retention of enantioselectivity and moderate yield could also be achieved when the migrating group was changed from p-tolyl to p-Br-phenyl group (**4u**). On gram-scale, the synthesis of product **4a** under the standard conditions proceeded in slightly deteriorated yield but with complete preservation of stereochemical integrity, demonstrating the potential utility in large-scale chemical synthesis.

Having demonstrated the efficacy of sulfur dioxide insertion with diazonium salts to induce asymmetric Truce–Smiles rearrangement with step economy, we sought to extend the reach of this strategy beyond the imitation imposed by the choices of aryldiazonium salt. To this end, the thianthrenium salts that could be readily accessed through the thianthrenation of arenes or alkyl alcohols[56,57] were evaluated for their compatibility as free radical precursors in the sulfur dioxide insertion mechanism. Regrettably, the previous conditions could not be directly translated to this class of reactants, yielding no desired product. A switch of photosensitizer from Mes-AcrClO$_4$ to fac-Ir(ppy)$_3$ favorably delivered the expected product **4a** in 43% yield. A more extensive examination of other reaction variables (see Supplementary Table 1) concluded the most optimal conditions used to survey reaction scope in Fig. 3: **1** (0.2 mmol), **3** (0.4 mmol), Rongalite (0.24 mmol), NaOH (0.24 mmol), fac-Ir(ppy)$_3$ (2 mol%), DABCO·(SO$_2$)$_2$ (0.4 mmol), MeCN (3.0 mL), blue LED, under N$_2$ at 20 °C for 18 h.

A variety of aryl thianthrenium salts could efficiently support the asymmetric rearrangement with commendable stereochemical control (**4**, 45–81%, 74–98% ee). Expectedly, several selected chiral sulfones derived with previous conditions were also readily furnished in this system with comparable yields and enantioselectivities (**4a**, **4b**, **4d**, **4i**, **4j**, **4r** and **4s**). 2,4-Dimethylphenyl substrate participated in the reaction with slightly diminished (**4aa**) enantioselectivity, perhaps due to steric congestion that hindered the stereochemical control during the arene transposition. Aryl thianthrenium salts bearing 3-cyano-4-isobutoxy or 4-phenoxy substituent gave sulfones **4ab** and **4ac**, respectively, with good results but the yield declined to 45% (**4ad**) when a 4-nitrophenoxy group was equipped in place of the alkoxy groups. Notably, amide functionalities were well tolerated in this reaction to furnish the corresponding products in excellent yields and enantioselectivities (**4ae** and **4af**). Heterocyclic dibenzofuran substrate was examined as well which provided product **4ag** in 60% yield and 96% ee. When the N-phenyl group in sulfinyl acrylamides was varied to N-Tol or N-Bn entity (**4ah** and **4ai**), the excellent reaction outcomes were largely unperturbed. As an intriguing complement to the diazonium salt system, the smooth participation of alkyl radical precursors including sensitive alkyl bromide notably expands the reaction scope despite the slightly lower enantioselectivity (**4aj**–**4am**). While the substrate scope has been explored by using p-tolyl as the migrating group, a switch to 4-Br-phenyl (**4an**) had no significant effect on yield and enantioselectivity of the Smiles rearrangement despite the change in electronics. By contrast, a more notable influence on the reaction was exhibited in modulating the $R^2$ group tethered to alkene. The less sterically hindered H substituent gave the product **4ao** in 45% yield and 94% ee, while the benzylic substrate could not be smoothly converted to **4ar**. As a testament to the robustness and chemoselectivity of this system to operate in elaborated setting, we demonstrated that famoxadone and estrone could be derivatized into sulfones (**4ap** and **4aq**) with β-quaternary stereocenter through the developed sulfur dioxide insertion and subsequent Truce–Smiles rearrangement process. This portends the applicability of this method in medicinal chemistry setting.

### Controlled experiments and proposed reaction processes

To gain insights into mechanistic details, a series of control experiments were carried out, as displayed in Fig. 4a. The transformation was suppressed completely when the radical scavenger, 2,2,6,6-tetramethylpiperidineoxy (TEMPO) was added to the standard conditions. Butylated hydroxytoluene (BHT) hampered the product yield of **3a** to 35% whereas the addition of 1,1-diphenylethylene gave rise to **4a** in trace amounts alongside (2-tosylethene-1,1-diyl)dibenzene **5** in 39% yield under standard conditions. These results converge on a radical pathway that is operative in current transformation. Subsequent studies showed that neither sodium dithionite nor aryldiazonium salt could achieve fluorescence quenching of photosensitizer Mes-AcrClO$_4$. Another important indication came from the production of **4a** in 50% yield when Mes-AcrClO$_4$ was removed from the standard conditions, implying that current radical sulfonylation process could proceed spontaneously without photosensitizers, albeit with decreased efficiency. Furthermore, a quantum yield of 3.28 determined for this reaction could elucidate the speculated spontaneity of this process.

These experimental data led to proposal of a plausible working mechanism for this photocatalytic asymmetric Smiles rearrangements based on diazonium salts, as shown in Fig. 4b. Initially, the homolytic cleavage of sodium dithionite generates two sulfur dioxide radical anions **A**, which could undergo single-electron transfer (SET) with aryldiazonium salt to release aryl radical **B** with simultaneous extrusion of SO$_2$. It is reasoned that the decomposition of sodium bisulfite could be an alternative source of SO$_2$ in this system, as inferred from the improved conversion upon the inclusion of this additive. This is rapidly ensued by insertion of aryl radical **B** into sulfur dioxide to give sulfonyl radical species **C** that adds to the double bond to form new radical intermediate **D**. The radical

**Fig. 2 | Generality of enantioselective Smiles rearrangement involving aryl-diazonium tetrafluoroborates.** Reaction conditions: **1** (0.2 mmol), **2** (0.4 mmol), Na$_2$S$_2$O$_4$ (0.4 mmol), Mes-AcrClO$_4$ (2 mol%), NaHSO$_3$ (0.3 mmol), MeCN (3.0 mL), 35 W blue LED, under N$_2$ at 10 °C for 18 h.

Truce–Smiles rearrangement proceeds spontaneously to generate the SO-centered radical **F**[54], traversing through a spirocyclic transition state **E** in an exothermic process[58]. The SO-centered radical **F** undergoes SET oxidation with aryldiazonium salt and further reacts with NaOH to convert into final product **4** and NaHSO$_3$. The regenerated NaHSO$_3$ and aryl radicals would enter the reaction to complete the catalytic cycle. By introducing a photocatalytic system, the photoexcited Mes-Acr$^{+*}$ could accelerate the oxidation of

intermediate **F** to product **4** through a SET process while the Mes-Acr· radical promotes the generation of aryl radical **B** from diazonium salt **2**, thereby improving the overall efficiency of this reaction process. Based on quantum yield of 2.43 and Stern–Volmer quenching experiments (for the experimental details, see Supplementary Methods), an analogous photoredox catalytic mechanism for the Ir(III) catalytic system has also been proposed and included in the Supplementary Fig. 6.

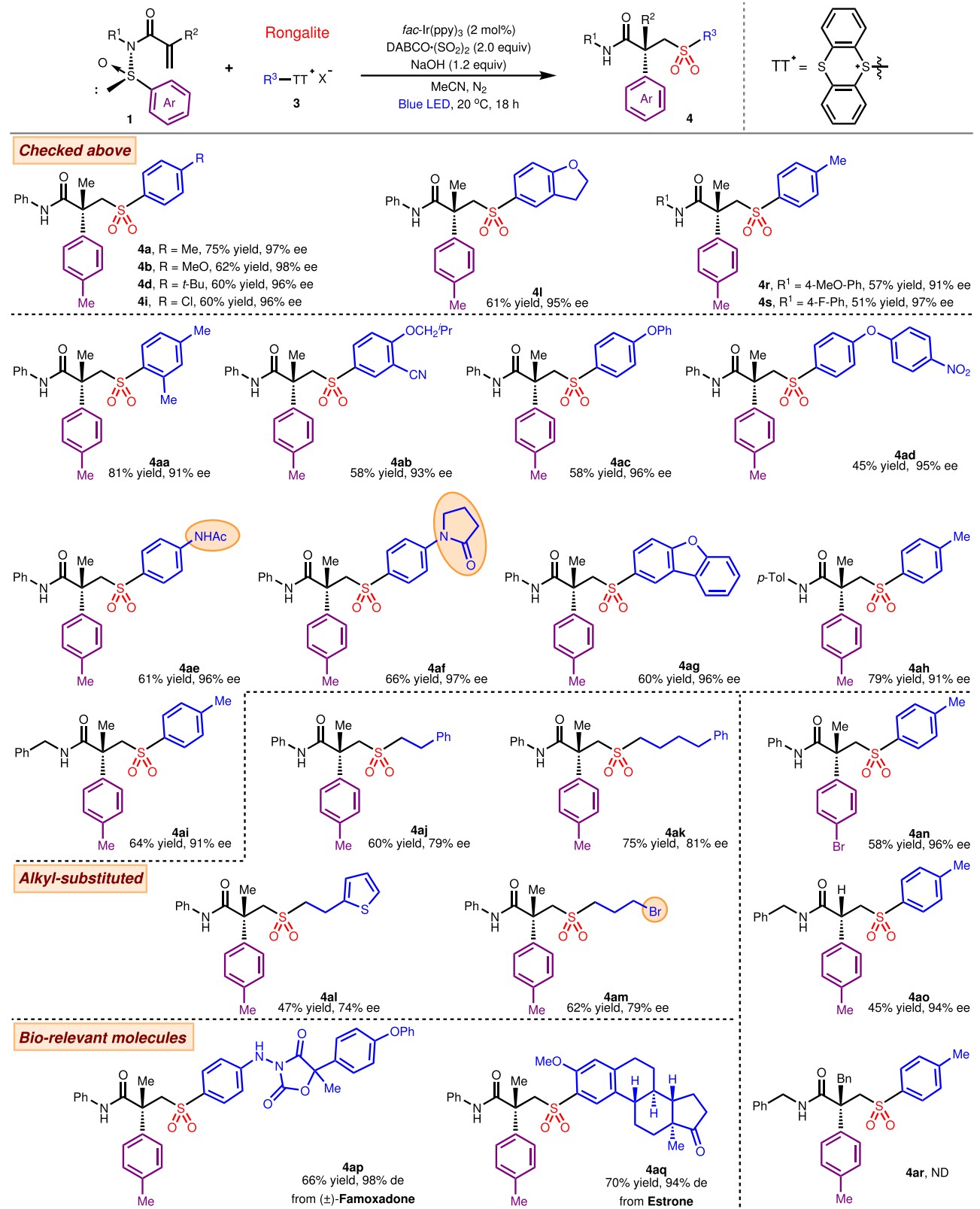

**Fig. 3 | Generality of enantioselective Smiles rearrangement involving thianthrenium salts.** Reaction conditions: **1** (0.2 mmol), **3** (0.4 mmol), Rongalite (0.24 mmol), NaOH (0.24 mmol), *fac*-Ir(ppy)₃ (2 mol%), DABCO·(SO₂)₂ (0.4 mmol), MeCN (3.0 mL), 35 W blue LED, under N₂ at 10 °C for 18 h.

## Discussion

In summary, we have developed an effective enantioselective Truce–Smiles rearrangement strategy under photoredox catalysis, enabling simultaneous construction of a stereodefined a

quaternary carbon center and the introduction of a sulfonyl group. The versatility of sulfur dioxide insertion strategy is exemplified in the compatibility of diazonium and thianthrenium salts to generate diverse functionalized sulfonyl radical for initiating the

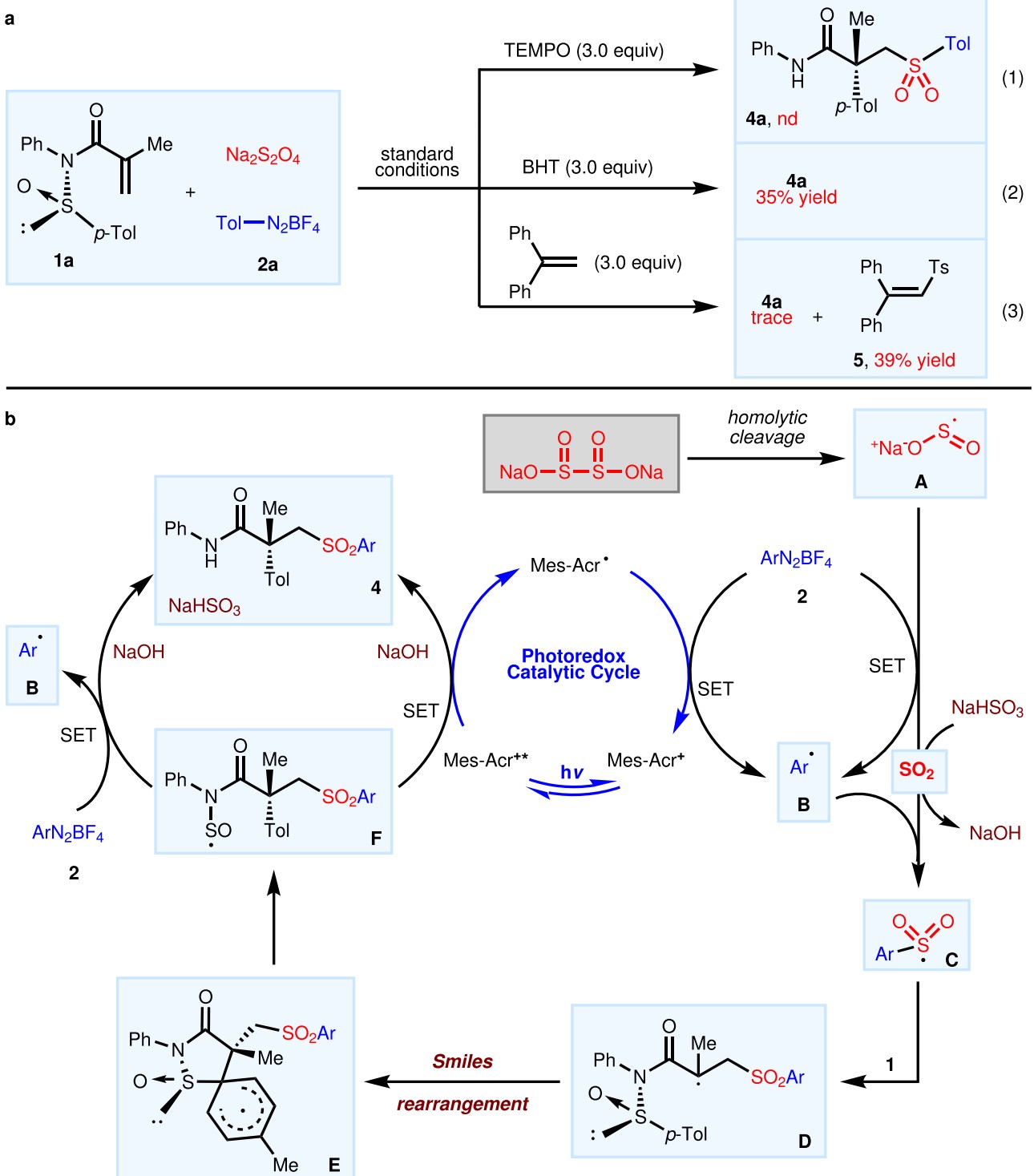

**Fig. 4 | Control experiments and plausible mechanism. a** Control experiments: (1) With TEMPO as a radical scavenger; (2) With BHT as a radical scavenger; (3) With 1,1-diphenylethylene as a radical scavenger. **b** Proposed mechanism for photocatalytic asymmetric Smiles rearrangements based on diazonium salts.

stereospecific rearrangement process. The transfer of chirality from sulfur in sulfoxide to the quaternary carbon is achieved during aryl migration and carbon-carbon bond formation. By in situ generation of radicals, this three-component coupling chemistry exhibits good functional group compatibility and step economy. The application to late-stage functionalization of bioactive molecules further demonstrates the practicality and potential of this strategy.

## Methods

### General procedure for the synthesis of 4

Method A: in a glove box, a dry quartz vial equipped with a magnetic stir bar is charged sequentially with **1a** (59.8 mg, 0.2 mmol), **2a** (82.4 mg, 0.4 mmol), $Na_2S_2O_4$ (69.6 mg, 0.4 mmol), $NaHSO_3$ (31.2 mg, 0.3 mmol), Mes-AcrClO$_4$ (1.6 mg, 2 mol%), and dry MeCN (3.0 mL). The reaction mixture is stirred for 18 h at 900 rpm in a thermostatic water bath at 20 °C under a 35 W blue LED light. When the reaction is

completed (monitored by TLC), the mixture is purified by flash chromatography on silica gel eluted with PE/EA (5/1) to afford the corresponding products. Method B: in a glove box, a dry quartz vial equipped with a magnetic stir bar is charged sequentially with **1a** (59.8 mg, 0.2 mmol), **3a** (182.4 mg, 0.4 mmol), Rongalite (37.2 mg, 0.24 mmol), DABCO·$(SO_2)_2$ (96.1 mg, 0.4 mmol), NaOH (9.6 mg, 0.24 mmol), *fac*-Ir(ppy)$_3$ (2.6 mg, 2 mol%) and dry MeCN (3.0 mL). The reaction mixture is stirred for 18 h at 900 rpm in a thermostatic water bath at 20 °C under a 35 W blue LED light. When the reaction is completed (monitored by TLC), the mixture is purified by flash chromatography on silica gel eluted with PE/EA (5/1) to afford the corresponding products.

## Data availability

Data relating to the materials and methods, optimization studies, experimental procedures, mechanistic studies, HPLC spectra, NMR spectra and mass spectrometry are available in the Supplementary information. The X-ray crystallographic coordinates for structures reported in this article have been deposited at the Cambridge Crystallographic Data Centre (CCDC), under deposition numbers CCDC 2208406. These data can be obtained free of charge from The Cambridge Crystallographic Data Centre via http://www.ccdc.cam.ac.uk/data_request/cif.

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

## Acknowledgements

We are grateful for financial support from National Natural Science Foundation of China (No. 22171206, J.W.), the Leading Innovative and Entrepreneur Team Introduction Program of Zhejiang (No. 2019R01005, J.W.), the Natural Science Foundation of Zhejiang Province (LY22B020003, S.L.), and the Open Research Fund of School of Chemistry and Chemical Engineering, Henan Normal University (2020ZD04, J.W.).

## Author contributions

J.W., S.L. and Z.C. conceived and directed the project. J.H. and F.L. designed and performed experiments and prepared the supplementary information. L.-H.Z. participated in the reaction mechanism discussion. S.L. wrote the paper. All authors commented on the manuscript.

## Competing interests

The authors declare no competing interests.
