## [Peer Review File · Nature Communications]

Accessing chiral sulfones bearing quaternary carbon stereocenters via photoinduced radical sulfur dioxide insertion and Truce-Smiles rearrangementREVIEWER COMMENTS

Reviewer #1 (Remarks to the Author):

In this manuscript, Wu et al. report a novel strategy based on photo-induced sulfur dioxide insertion to trigger radical-type asymmetric Truce-Smiles rearrangements. The three-component reactions of an N-arylsulfinyl acrylamide, a SO₂ surrogate (Na₂S₂O₄ or Rongalite) and an aryl radical precursor (diazonium salts or aryl thianthrenium salts) proceed smoothly under mild conditions, furnishing a wide variety of biologically and synthetically valuable sulfone derivatives bearing a quaternary carbon stereocenter in good yields and with high enantiomeric excess. The mechanistic investigations are insightful and the proposed mechanism looks reasonable. Overall, it is a potentially important contribution to organic synthesis, photocatalysis and green chemistry. I support its publication in Nature Communications after minor revisions.

Minor points:

(1) In some sulfone dioxide insertion reactions, different SO₂ surrogates such as DABCO·(SO₂)₂ often lead to different reactivity and/or selectivity, even in enantioselective reactions. How about their performance in this asymmetric Truce-Smiles rearrangement reaction?

(2) In table 1, the authors have examined the reaction in the absence of visible light at room temperature. How about heating it at higher temperature (70 degrees) without light irradiation?

(3) HPLC spectra of 1d and 4s (method A) show signals with uncommon shape, that of 4f has low resolution. These might lead to inaccurate determination of enantiomeric excess. NMR spectra of some compounds, such as 4e, 4g, 4r, 4aa and 4ac, contain obvious impurities. Please improve them and update the yields and spectra.

(4) The reaction set-up and emission spectra of the light source, in particular λ_{max} , should be provided so that the readers can easily follow the procedure.

(5) Please check through the main text and ESI, there are some minor typos errors and inconsistency.

For example, the light source is described as 'blue LED' while sometimes as 'blue LEDs', the photocatalyst as 'MesAcr+' while sometimes as 'MesAcr+ClO₄'.

In table 1, entry 17, the photocatalyst is termed in an incorrect way.

In Fig 3, some characters and symbols are messed up.

In the conclusion section, one sentence "The transfer of S-chirality to central chirality is perfectly achieved during aryl migration and formation of quaternary carbon." is confusing, both sulfur- and carbon-centered chirality are central chirality.

Reviewer #2 (Remarks to the Author):

In this manuscript, Wu and co-workers report the enantioselective synthesis of chiral sulfones bearing quaternary carbon stereocenters through a radical sulfur dioxide insertion triggered asymmetric Smiles rearrangement strategy. On the basis of above tactics, the authors present two practical photoredox catalytic protocols, with aryl diazonium salts and thianthrenium salts as radical precursors respectively, to generate in situ sulfonyl radicals for initiating the stereospecific rearrangement process. By contrast with Nevado's seminal work, this research could detour the tedious preparation and the poor functional group compatibility of sulfonyl chlorides, thereby conferring this rearrangement reaction broader application potential. Moreover, the intriguing SO₂ insertion solutions and unusual photoredox catalytic cycle mechanism demonstrated in this transformation enable this study more distinctive and attractive in chemical nature. Although investigation on functional group applicability and reaction mechanism is slightly insufficient, this is a nice work and I recommend the publication of this manuscript in Nature Communications after following modifications being

addressed:

(1) As regarding to substrate1, groups tethered to alkenes and migrating aryl groups are critical for stereochemical control. However, only Me and p-tol were investigated in this manuscript, how about other alkyl and aryl groups for this transformation?

(2) To gain more insight for this reaction, quantum yield and Stern-Volmer quenching experiments are suggested to be performed.

(3) The relative references regarding radical aryl migration should cited: such as Science Advances 2017, 3, e1701487; Chem. Commun. 2017, 53, 4038; Nat. Commun. 2016, 7, 13852.

Reviewer #3 (Remarks to the Author):

As requested by the editor, this review is concerned solely with the crystal structure that accompanies manuscript NCOMMS-22-25475.

From the checkCIF report provided, the structure appears to be of a publishable standard. The refinement statistics are quite typical of many room temperature structures published nowadays. There is sufficient anomalous signal from the S and Cl atoms to assign the absolute structure [Flack $x(u) = 0.02(2)$ as presented]. For all intents and purposes, the structure is justified for inclusion in the paper.

To analyze the structure in more detail, I extracted the dataset (hkl file) and model (res file) from the provided CIF. Subsequent refinement revealed a few minor issues that are easily fixed and that result in a higher quality structure using more of the original data.

In the original structure refinement, the authors have truncated the high-resolution limit at 50° in 2θ (statement OMIT 0.00 50.00). In addition, they have omitted a further six reflections using OMIT statements [reflections (020), (200), (022), (242), (-242), (203)]. On close inspection, only one of these (020) can justifiably be omitted (due to it being obscured by the beamstop). The other reflections can and should be included.

On refining to convergence with a higher resolution cutoff (OMIT -3 55) and including all but (020), a difference map revealed a few small electron density peaks that were consistent with a second position of the SO₂ group. This would in turn require a second orientation of the chlorinated benzene ring. I constructed a disorder model with suitable constraints/restraints. Refinement of this disordered model is dramatically superior to the original. I would recommend that the authors either use this revised model (a res file is attached) or construct their own.

Specific questions suggested on the manuscript review site:

- What are the noteworthy results?

This review concerns only the crystal structure presented. The model reveals the chemical structure well enough, but could easily be improved.

- Will the work be of significance to the field and related fields? How does it compare to the established literature? If the work is not original, please provide relevant references.

The quality of the structure is similar to that of many published organic crystal structures. Nevertheless, it could easily be improved.

- Does the work support the conclusions and claims, or is additional evidence needed?

It is adequate.

- Are there any flaws in the data analysis, interpretation and conclusions? - Do these prohibit

publication or require revision?

There are a few minor flaws in the execution of the structure refinement, but they are easily fixed and result in a much improved structure.

- Is the methodology sound? Does the work meet the expected standards in your field?

It is similar quality to many published structures, but as stated above, it could easily be improved.

- Is there enough detail provided in the methods for the work to be reproduced?

Yes.

Our Responses to the Comments of Referees

Comments of Referee #1

In this manuscript, Wu et al. report a novel strategy based on photo-induced sulfur dioxide insertion to trigger radical-type asymmetric Truce-Smiles rearrangements. The three-component reactions of an N-arylsulfinyl acrylamide, a SO₂ surrogate (Na₂S₂O₄ or Rongalite) and an aryl radical precursor (diazonium salts or aryl thianthrenium salts) proceed smoothly under mild conditions, furnishing a wide variety of biologically and synthetically valuable sulfone derivatives bearing a quaternary carbon stereocenter in good yields and with high enantiomeric excess. The mechanistic investigations are insightful and the proposed mechanism looks reasonable. Overall, it is a potentially important contribution to organic synthesis, photocatalysis and green chemistry. I support its publication in Nature Communications after minor revisions.

Our response: We are very grateful for the positive remarks that the reviewer has had on this work. We appreciate the efforts and time devoted to review this manuscript.

(1) *In some sulfone dioxide insertion reactions, different SO₂ surrogates such as DABCO·(SO₂)₂ often lead to different reactivity and/or selectivity, even in enantioselective reactions. How about their performance in this asymmetric Truce-Smiles rearrangement reaction?*

Our response: We thank the referee for raising this question. In fact, we have examined various SO₂ surrogates including DABCO·(SO₂)₂, Na₂S₂O₅, K₂S₂O₅, Na₂SO₃, NaHSO₃ and Na₂S₂O₄ at the beginning of this study. Na₂S₂O₄ was found to provide the best overall reaction outcomes with respect to product yield and ee. The screening results have been summarized in the table below.

Entry	"SO ₂ " source	PC	Yield (%)	Ee (%)
1	DABCO·(SO ₂) ₂	Mes-AcrClO ₄	47	94
2	Na ₂ S ₂ O ₅	Mes-AcrClO ₄	40	93
3	K ₂ S ₂ O ₅	Mes-AcrClO ₄	43	94
4	Na ₂ SO ₃	Mes-AcrClO ₄	35	93
5	NaHSO ₃	Mes-AcrClO ₄	31	93
6	Na ₂ S ₂ O ₄	Mes-AcrClO ₄	54	94

Conditions: **1a** (0.2 mmol), **2a** (0.4 mmol), "SO₂" source (0.4 mmol), photocatalyst (2 mol%), additive (1.5 equiv), solvent (3.0 mL), 35 W blue LED, under N₂ at 20 °C for 18 h.

(2) In table 1, the authors have examined the reaction in the absence of visible light at room temperature. How about heating it at higher temperature (70 degrees) without light irradiation?

Our response: When the model reaction was carried out at 70°C in the absence of visible light, the product 4a was obtained in 42% yield, indicating that light irradiation is crucial for the transformation.

(3) HPLC spectra of 1d and 4s (method A) show signals with uncommon shape, that of 4f has low resolution. These might lead to inaccurate determination of enantiomeric excess. NMR spectra of some compounds, such as 4e, 4g, 4r, 4aa and 4ac, contain obvious impurities. Please improve them and update the yields and spectra.

Our response: In the revised supporting information, the HPLC spectra of **1d**, **4s** (method A) and **4f** have been updated after the analyses have been performed with a new method. Additionally, we have recorded new ¹H NMR and ¹³C NMR for compounds **4e**, **4g**, **4r**, **4aa** and **4ac** after complete removal of solvent residue and impurities. The clean spectra and yields have been updated in the revised supporting information and manuscript.

(4) The reaction set-up and emission spectra of the light source, in particular λ_{max} , should be provided so that the readers can easily follow the procedure.

Our response: In accord with the suggestion of referee, the device and related parameters used in this photocatalytic system have been detailed in the revised supporting information.

(5) Please check through the main text and ESI, there are some minor typos errors and inconsistency.

For example, the light source is described as 'blue LED' while sometimes as 'blue LEDs', the photocatalyst as 'MesAcr+' while sometimes as 'MesAcr+ClO4-'.

In table 1, entry 17, the photocatalyst is termed in an incorrect way.

In Fig 3, some characters and symbols are messed up.

Our response: We appreciate the efforts of the reviewer to carefully check and pinpoint these errors and inconsistency. Throughout the manuscript, the light source 'blue LEDs' and photosensitizer 'MesAcr⁺ or MesAcr⁺ClO₄⁻' are standardized as 'blue LED' and 'Mes-AcrClO₄', respectively. In table 1, entry 17, The photocatalyst 'MesAcr⁺' is corrected to 'Mes-AcrClO₄'. In Fig 3, all structures have been redrawn and checked carefully.

In the conclusion section, one sentence "The transfer of S-chirality to central chirality is perfectly achieved during aryl migration and formation of quaternary carbon." is confusing, both sulfur- and carbon-centered chirality are central chirality.

Our response: We have revised this confusing expression based on the reviewer's suggestion. The sentence "The transfer of S-chirality to central chirality is perfectly achieved during aryl migration and formation of quaternary carbon." has been rephrased to 'The transfer of chirality from sulfur in sulfoxide to the quaternary carbon

is perfectly achieved during aryl migration and carbon-carbon bond formation.'.

Comments of Referee #2

In this manuscript, Wu and co-workers report the enantioselective synthesis of chiral sulfones bearing quaternary carbon stereocenters through a radical sulfur dioxide insertion triggered asymmetric Smiles rearrangement strategy. On the basis of above tactics, the authors present two practical photoredox catalytic protocols, with aryl diazonium salts and thianthrenium salts as radical precursors respectively, to generate in situ sulfonyl radicals for initiating the stereospecific rearrangement process. By contrast with Nevado's seminal work, this research could detour the tedious preparation and the poor functional group compatibility of sulfonyl chlorides, thereby conferring this rearrangement reaction broader application potential. Moreover, the intriguing SO₂ insertion solutions and unusual photoredox catalytic cycle mechanism demonstrated in this transformation enable this study more distinctive and attractive in chemical nature. Although investigation on functional group applicability and reaction mechanism is slightly insufficient, this is a nice work and I recommend the publication of this manuscript in Nature Communications after following modifications being addressed.

Our response: We thank the referee for supporting the publication of this manuscript in Nature Communications. The suggestions have been looked into carefully to refine the manuscript.

(1) As regarding to substrate 1, groups tethered to alkenes and migrating aryl groups are critical for stereochemical control. However, only Me and p-tol were investigated in this manuscript, how about other alkyl and aryl groups for this transformation?

Our response: We appreciate the valuable suggestions of the referee and have conducted studies to enrich the range of substrates. As shown in table below, substrate **1e** that carries a *p*-bromo aryl substituent was successfully converted in both catalytic systems. The formation of products **4u** and **4an** in high enantioselectivity and moderate yield indicate that the migratory group had slight effect on the efficiency of conversion. In contrast, the modulation of R² group tethered to alkene exhibits a more substantial effect on the transformation. The less sterically hindered H group gave the product **4ao** in 45% yield and 94% ee, while the benzylic substrate **1g** could not be smoothly converted to **4ar** under various reaction conditions. We have included these results in the modified manuscript and supporting information. Due to these additions, the compounds originally represented by **4an** and **4ao** have been relabeled as **4ap** and **4aq**, respectively.

(2) To gain more insight for this reaction, quantum yield and Stern-Volmer quenching experiments are suggested to be performed.

Our response: In accord with the suggestion of referee, the quantum yield and Stern-Volmer quenching experiments have been carried out. The results are summarized as follows:

- 1) With regard to aryldiazonium salt system, the quantum yield is determined to be 3.28. This is consistent with the spontaneous reaction phenomenon existing in the system and corroborates well with our proposed mechanism.
- 2) For the thianthrenium salt system, Stern-Volmer quenching experiments indicated that substrate 3a could effectively quench the excited $^*Ir(ppy)_3$ but 1c could not. A quantum yield of 2.43 was measured, which revealed that the extended radical-chain reactions were possible. According to these data, we made the corresponding corrections on the initially proposed mechanism. More detailed experimental and mechanistic descriptions are provided in the revised supporting information.

(3) The relative references regarding radical aryl migration should be cited: such as *Science Advances* 2017, 3, e1701487; *Chem. Commun.* 2017, 53, 4038; *Nat. Commun.* 2016, 7, 13852.

Our response: We thank the referee for highlighting the relevant works on radical aryl migration to us, which have been cited as *Refs 40-42* in the revised manuscript.

Comments of Referee #3

As requested by the editor, this review is concerned solely with the crystal structure that accompanies manuscript NCOMMS-22-25475.

From the checkCIF report provided, the structure appears to be of a publishable standard. The refinement statistics are quite typical of many room temperature structures published nowadays. There is sufficient anomalous signal from the S and Cl atoms to assign the absolute structure [Flack $x(u) = 0.02(2)$ as presented]. For all intents and purposes, the structure is justified for inclusion in the paper.

To analyze the structure in more detail, I extracted the dataset (hkl file) and model (res file) from the provided CIF. Subsequent refinement revealed a few minor issues that are easily fixed and that result in a higher quality structure using more of the original data.

In the original structure refinement, the authors have truncated the high-resolution limit at 50° in 2theta (statement OMIT 0.00 50.00). In addition, they have omitted a further six reflections using OMIT statements [reflections (020), (200), (022), (242), (-242), (203)]. On close inspection, only one of these (020) can justifiably be omitted (due to it being obscured by the beamstop). The other reflections can and should be included.

On refining to convergence with a higher resolution cutoff (OMIT -3 55) and including all but (020), a difference map revealed a few small electron density peaks that were consistent with a second position of the SO₂ group. This would in turn require a second orientation of the chlorinated benzene ring. I constructed a disorder model with suitable constraints/restraints. Refinement of this disordered model is dramatically superior to the original. I would recommend that the authors either use this revised model (a res file is attached) or construct their own.

Specific questions suggested on the manuscript review site:

- *What are the noteworthy results?*

This review concerns only the crystal structure presented. The model reveals the chemical structure well enough, but could easily be improved.

- *Will the work be of significance to the field and related fields? How does it compare to the established literature? If the work is not original, please provide relevant references.*

The quality of the structure is similar to that of many published organic crystal structures. Nevertheless, it could easily be improved.

- *Does the work support the conclusions and claims, or is additional evidence needed?*

It is adequate.

- *Are there any flaws in the data analysis, interpretation and conclusions? - Do these prohibit publication or require revision?*

There are a few minor flaws in the execution of the structure refinement, but they are easily fixed and result in a much improved structure.

- *Is the methodology sound? Does the work meet the expected standards in your field?*

It is similar quality to many published structures, but as stated above, it could easily be improved.

- *Is there enough detail provided in the methods for the work to be reproduced?*

Yes.

Our response: We thank the referee for the meticulous check on the crystal structure reported in this manuscript. We have thus refined the crystal structure based on the model provided by the reviewer. The improved data has been re-deposited at the Cambridge Crystallographic Data Centre (CCDC) under the reassigned CCDC number 2208406.

In summary, we have completed a series of experiments and carefully revised the manuscript according to the reviewers' suggestions. The new results have been included in the revised manuscript and Supporting Information. The compound number in Figure 2 and 3 of the manuscript as well as those in supporting information have been updated accordingly.

REVIEWERS' COMMENTS

Reviewer #1 (Remarks to the Author):

The authors have carefully addressed all the questions and concerns raised by the reviewers. I'm satisfied with the revised version and recommend its publication in Nat. Commun. as it is.

Reviewer #2 (Remarks to the Author):

Authors have addressed all of my concerns in this revised version. I recommend accepting the manuscript without any revisions.